# Non-Coding RNAs in Human Cancer and Other Diseases: Overview of the Diagnostic Potential

**DOI:** 10.3390/ijms242216213

**Published:** 2023-11-11

**Authors:** Roman Beňačka, Daniela Szabóová, Zuzana Guľašová, Zdenka Hertelyová, Jozef Radoňak

**Affiliations:** 1Department of Pathophysiology, Medical Faculty, Pavol Jozef Šafarik University, 04011 Košice, Slovakia; roman.benacka@upjs.sk (R.B.); daniela.szaboova@student.upjs.sk (D.S.); 2Center of Clinical and Preclinical Research MEDIPARK, Pavol Jozef Šafarik University, 04011 Košice, Slovakia; zuzana.gulasova@upjs.sk (Z.G.); zdenka.hertelyova@upjs.sk (Z.H.); 31st Department of Surgery, Faculty of Medicine, Louis Pasteur University Hospital (UNLP) and Pavol Jozef Šafarik University, 04011 Košice, Slovakia

**Keywords:** non-coding RNAs, microRNA, small interfering RNA, piwi-interacting RNA, long non-coding RNA, small nuclear RNA, small nucleolar RNA, cancer, biomarkers

## Abstract

Non-coding RNAs (ncRNAs) are abundant single-stranded RNA molecules in human cells, involved in various cellular processes ranging from DNA replication and mRNA translation regulation to genome stability defense. MicroRNAs are multifunctional ncRNA molecules of 18–24 nt in length, involved in gene silencing through base-pair complementary binding to target mRNA transcripts. piwi-interacting RNAs are an animal-specific class of small ncRNAs sized 26–31 nt, responsible for the defense of genome stability via the epigenetic and post-transcriptional silencing of transposable elements. Long non-coding RNAs are ncRNA molecules defined as transcripts of more than 200 nucleotides, their function depending on localization, and varying from the regulation of cell differentiation and development to the regulation of telomere-specific heterochromatin modifications. The current review provides recent data on the several forms of small and long non-coding RNA’s potential to act as diagnostic, prognostic or therapeutic target for various human diseases.

## 1. Introduction

The Encyclopedia of DNA Elements (ENCODE) project indicates that most of the human genome is transcribed and consists of app. 60,000 genes, of which app. 1/3 (app. 20,000 genes) encode proteins and approximately 16,000 are protein non-coding RNAs and 14,000 pseudogenes [1]. It is estimated that at least 80% of the human genome is biologically active, i.e., transcribed. In contrast to this “transcriptional activity”, only a small part of human DNA (structural genes) encodes for proteins (2%); the rest (98%) is non-coding DNA and for a long time was considered junk, evolutionarily accumulated reserve copies, supplementary matter, etc. [2]. Recently, the role of this puzzling part of the genome started to be disentangled as the gigantic world of ncRNAs was disclosed. Non-coding RNAs (ncRNAs) are RNA molecules encoded by RNA genes, which are ultimately not translated into proteins [3,4]. Non-coding RNAs can be encoded by their own genes (RNA genes) in areas in between structural genes (intergenic ncRNA), but can also be derived from protein-coding genes, i.e., intronic or even exonic parts of mRNA, which are further spliced or modified [5,6]. Their involvement in cellular processes is very varied, including gene regulation, translation, RNA splicing [7], DNA replication [8], genome defense, and chromosome structure [9]. They participate in virtually all aspects of genetic and epigenetic regulations [5,10] and are of enormous interest due to their use in practical medicine.

According to the size of the chain, ncRNAs are subdivided into (i) *short non-coding RNA* (*sncRNA*) containing less than 200 nucleotides (nt) and (ii) *long non-coding RNAs* (*lncRNA*) that contain more than 200 nt. The first group comprises several subgroups, with prominent examples being *transfer RNA* (*tRNA*) and *ribosomal RNA* (*rRNA*), and less known examples include small RNAs such as *micro RNAs* (*miRNA*), *small interfering RNA* (*siRNA*), *piwi-interacting RNA* (*piRNA*), *small nuclear RNA* (*snRNA*), *small nucleolar RNAs* (*snoRNAs*), *extracellular RNA* (*exRNA*), and *small Cajal body-specific RNAs* (*scaRNAs*). The second group, *long ncRNAs*, are subdivided based on their roles in the cell—biogenesis (intronic RNA, enhancer RNA, promoter RNA, antisense RNA, sense RNA, intergenic RNA, bidirectional RNA), structure (linear lncRNA, circular lncRNA, long intergenic non-coding RNA, enhancer-derived RNA, transcribed ultra-conserved RNA, natural antisense transcript), and action (cis-acting long non-coding RNA, competing endogenous RNA, trans-acting long non-coding RNA) [5]. With respect to function, ncRNAs can generally be divided into two main classes—*structural ncRNAs* and *regulatory ncRNAs*. The structural ncRNA group includes the abovementioned *rRNAs*, *tRNA*, *snRNA*, and *snoRNA*, as well as *small cytoplasmic RNA*, and *promoter upstream transcripts* (*PROMPTs*). Regulatory ncRNAs can be further subdivided into: (i) *small non-coding RNA (miRNA, piRNA, siRNA, centromere repeat-associated small interacting RNA, telomere-specific small RNA)*, (ii) *medium-sized ncRNAs*, and (iii) *long ncRNAs* [11].

In this review, we discuss the role of ncRNAs and their diagnostic and prognostic use in human diseases, including various forms of cancers [12,13,14,15]. Some of them, e.g., circRNA and siRNAs, appear to also be promising therapeutical tools and have been reviewed in excellent ways in recent works [16,17,18]. In the beginning of this work, we highlighted the main features and functions and clinical applications of selected ncRNA species in general, while in the second part we focus on particular body systems and their related disorders.

## 2. General Overview of ncRNA Biogenesis and Action

### 2.1. MicroRNAs

*MicroRNAs* (*miRNAs*) are a class of small, non-coding, single-stranded RNAs (18–24 nt) involved in the regulation of several cellular processes by repressing the translation of targeted mRNAs. The biogenesis of microRNAs starts in the nucleus via transcription from DNA sequences often located in the non-coding regions of the genome—introns—often near the sequences of mRNA that are subsequently affected by a given miRNA. The products of transcription are long primary transcripts with a characteristic hairpin structure (pri-miRNA), which is recognized and cleaved by RNAase III Drosha. This cleaving generates 70-nucleotide-long pre-microRNA molecules, which are actively exported (Ran-GTP, Ecportin 5) into the cytoplasm. In the cytoplasm, pre-microRNAs are cleaved by the RNAse III Dicer into double-stranded RNA molecules. An enzymatic cleavage by helicase follows, and final single-stranded microRNAs and antisense microRNAs are formed (Figure 1). The final single-stranded microRNA can be then incorporated into three possible complexes—miRISC (miRNA-containing RNA-induced silencing complex), miRNP (miRNA-containing ribonucleoprotein complex), or miRgonaute, and functions as a gene regulator at the post-transcriptional level [19,20].

***Function***. miRNAs regulate cellular processes via RNA silencing and the post-transcriptional regulation of gene expression through base-pair complementarity to target sequences and, in case of full complementarity, either cleaving the mRNA transcripts into two-strand pieces or destabilizing them by shortening their poly(A) tail. In case of incomplete complementarity, the entire RISC complex binds to the mRNA transcript that is inaccessible for translation [21].

***Clinical application***. Based on their cellular functions, the dysregulation of miRNA expression has been associated with a variety of human diseases. Many miRNAs are capable of directly inhibiting cell cycle genes and controlling cell proliferation, acting either as onco-miRNAs or tumour-suppressor miRNA; therefore, they have been closely connected to various human cancers as diagnostic/prognostic biomarkers or therapeutic agents. Historically, the first human disease associated with miRNA dysregulation was chronic lymphocytic leukemia, but this dysregulation has been since connected to more [22]. In many cancers, miRNAs are the causal factor behind increased cell proliferation, causing DNA repair deficiencies. In colorectal cancers, miR-155 has been observed to epigenetically methylate the *MLH1* gene, causing reduced DNA mismatch repair [23]. In hepatocellular carcinoma, miRNA miR-21 interacts with *MAP2K3*, promoting cell proliferation and giving rise to tumour mass [24]. In 28% of glioblastomas, an increased expression of miR-181b has been connected to a reduced expression of MGMT, a DNA repair enzyme, and a direct target of miR181b [25]. Aberrant levels of miRNAs have been observed in cardiovascular diseases like heart failure, acute myocardial infarction, pulmonary hypertension, or arrhythmias. miRNAs have been linked with atrial remodeling via the regulation of Ca^2+^ channel protein expression, specifically miR-328 and its direct targets: cardiac L-type Ca^2+^ channel α1c- and β1 subunits CACNA1C and CACNB2. Therefore, they affect atrial fibrillations [26]. miRNAs play a vital role in the control of genes significant to addiction. In the nucleus accumbens, a region of the basal forebrain responsible for controlling the sensations of reward that fuel motivational routines, miR-382, was shown to be downregulated. Dopamine receptor D1 (DRD1) is the target of miR-382, whose overexpression upregulates DRD1 and delta fosB (transcription factor), which starts a cascade of transcription events in the nucleus accumbens that eventually leads to addictive behavior [27].

### 2.2. Small Interfering RNAs

*Small interfering RNA* (*siRNA*), also called *short interfering RNA* or *silencing RNA*, is a class of *double-stranded RNA*, typically of 20–24 bp in length, with phosphorylated 5′ ends and hydroxylated 3′ ends with two overhanging nucleotides. They operate similarly to miRNA within the *RNA interference* (*RNAi*) *pathway*. However, in contrast to miRNA, they are fully complementary to mRNA, which is targeted. siRNAs are created within the nucleus from long dsRNA (either small hairpin RNAs or complementary RNAs), although siRNAs can also be introduced into cells through transfection. Similar to miRNA, molecules of siRNA bind to specific nucleoproteins (RNPs) from the Argonaut Ago subfamily to build up an *RNA-induced silencing complex* (*RISC*) (Figure 1). Longer ds pre-siRNA precursors (from 30 to over 100 nucleotides) can be cleaved by an endo-ribonuclease type III Dicer (as a part of the RISC complex) to the smaller final ds RNA. Once the double siRNA is part of the RISC complex, the siRNA is unwrapped to form *single-stranded siRNA*. The siRNA, as a part of the RISC complex, can scan around and bind to a complementary mRNA sequence of specific genes. The binding of siRNA-RNP to target mRNA induces mRNA cleavage as a posttranscriptional regulatory mechanism, preventing translation into amino acids and proteins [28].

***Function*.** Similar to miRNA, the major role of siRNA is the *silencing of gene expression using the interfering pathway to inactivate mRNA* and to repress translation. On the contrary, miRNAs have an incomparably bigger repertoire, broader specificity and paralleled actions compared to siRNA. While siRNA very precisely targets its specific mRNA (100% complementarity), miRNA can have multiple targets (possibly over 100 mRNAs at the same time). If the main action of siRNA is the endonucleolytic cleavage of targeted mRNA, in the case of miRNA, endonucleocytic cleavage is rare (in the case of really high complementarity). Rather, miRNA delays and represses translation through mRNA degradation or physical binding to a target mRNA [28].

***Clinical application*.** While miRNAs are more widely used as diagnostic and biomarker tools, siRNAS are promising therapeutical agents. The direct application of *synthetic siRNAs* opens novel possibilities in innovative therapies for human diseases in the post-genomic era. The first siRNA drugs received approval for clinical use by the US Food and Drug Administration and the European Medicines Agency between 2018 and 2022. In addition, research regarding siRNA usage in therapy helps improve the clinical applications of other molecular methodologies. This research includes siRNA dosages in conjugates, exact sequencing, suitable vectors of delivery to intended site to avoid unwanted target-off effects, and gene knockout applications, which include metabolic diseases (hypercholesterolemia, porphyrias), hematology, infectious diseases, oncology, and ocular diseases. [29,30,31]. A promising means of the intracellular delivery of siRNAs is via nano vectors, nano-scale delivery vehicles, capable of protecting siRNAs against degradation and facilitating the cellular uptake of siRNA. Currently, lipid-based nano vectors are showing great therapeutic potential for central nervous system disorders, as they are able to completely cross the blood–brain barrier [32].

### 2.3. Circular ncRNAs

*Circular ncRNAs* (*circRNAs*) are single-stranded regulatory ncRNAs with a closed circular structure (ring shape) formed by covalent bonds, which renders them highly stable against exonuclease-mediated degradation [33,34]. They are ubiquitous across species but not highly conserved among different species ranging from viruses to mammals. Most circRNAs (84%) are created from protein-coding genes by an alternative splicing called backsplicing of pre-mRNAs, connecting the 3′splice end of the downstream exon to the 5′splice site of the upstream exon of genes containing long exons and flanking introns [35,36]. circRNA can be transported from the nucleus to the cytosol by specific mechanisms. In human cells, nuclear RNA helicase UAP56 exports long circRNAs (>1200 nucleotides), while nuclear spliceosome RNA helicase URH49 exports short circRNAs (<400 nucleotides) in size-dependent manner [37,38].

***Function.*** CircRNAs can be implicated in the following important roles: (a) they can regulate protein synthesis and (b) they serve as posttranscriptional gene regulators through acting as so-called “*miRNA sponges*”. CircRNAs prevent miRNAs from binding to their target mRNAs, thus serving as competing endogenous miRNAs inhibitors, e.g., *CDR1AS* (*ciRS-7*) contains more than 70 miR-7 target sites and serves as a miRNA sponge in specific tissues [33,39]. *CircHIPK3* also sponges multiple miRNAs; it binds to tumour-suppressive miR-124, and overexpression of this circRNA promotes tumours [37].

***Clinical applications***. CircRNAs are expressed in human tissues or present with developmental stage-specific expression. Furthermore, they are very stable and resistant to degradation by nucleases, as seen in the nervous tissue [33,34]. Because of the specific expression and long-term stability of circRNAs, they can be used as potential biomarkers in tumour diagnostics (e.g., subgroupings of medulloblastoma based on circRMST) or gastric cancer based on hsa_circRNA_102958 [39]. Recent cumulative data examine the usage of circRNAs in breast cancer, lung cancer, colorectal cancer, kidney or bladder carcinomas, gastric cancer, glioma, hepatocellular carcinoma, pancreatic cancer, and leukemia [33]. Dysregulated circRNAs play important roles in various pathological conditions, including cancer, neurodegenerative diseases, osteoarthritis, diabetes, and cardiovascular diseases. [33,38]. CircRNA can be involved in cancer pathogenesis in several ways: (a) acting through the miRNA sponge, circRNAs may serve as competitive inhibitors of miRNA and behave as *oncogenes* (e.g., circCDDC66 in colon cancer, circCCAC1 in cholangiocarcinoma, WHSC1 in endometrial and ovarian cancer, circANRIL in breast, bladder and gastric cancers, and circRNA_0000285 in cervical cancer [35]). circPCNXL2 and hsa_circ_001895 cause kidney cancer by inhibiting miR-153 and miR-296-5p, respectively. circRNA hsa_circ_001783 in breast cancer acts through the miR-20c-3p sponge. circRNAs such as circHMGB2, circIGF2BP3 and circUSP7 promote the development of non-small cell carcinoma [35,40]. (b) Acting through the miRNA sponge pathway or through other mechanisms, circRNAs may behave as *tumour suppressors* (e.g., circHIPK3 in bladder cancer, circURI1 and hsa_circ_0014717, circORC5 in gastric cancer, circ-AKT3 in kidney cancer through the miR-296-3p sponge, and circPLCE1 in colon cancer) [33,39,40]. circRNA isolated from various body fluids can serve as possible *diagnostic* or *prognostic markers*, determining different subtypes or different staging and grading of the tumours (e.g., circRMST from blood in medulloblastoma, hsa_circ_0014717, hsa_circ_0000190 or hsa_circRNA_102958 from blood or from stomach samples in gastric cancer, hsa_circ_0001874, and hsa_circ_0001971 from local samples in oral carcinoma, circPDLIM5, circSCAF8, circPLXDC2, circSCAMP1, and circCCNT2 collected from urine or blood in prostate cancer) [33,35,38,39,40].

### 2.4. Piwi-Interacting ncRNAs

*PIWI-interacting RNAs* (*piRNAs*) are single-stranded ncRNAs (26–31 nucleotides in length) with a non-conserved and very diverse set of nucleotide sequences when compared with any other known cellular RNA family. piRNAs were first discovered in germ cells (2006), and now they constitute one of the largest known classes of ncRNA [15]. On the basis of their origin, piRNAs are divided into: (i) transposon-derived piRNA, (ii) lncRNA-derived piRNAs, and (iii) mRNA-derived piRNA. Only the function of the first group is well understood. The vast majority of piRNAs are clustered in relatively short genomic loci on chromosomes 17, 5, 4 and 2. piRNAs received their name due to their interaction with P-element-induced wimpy testis (PIWI) proteins, a germ-line-specific Argonaute family of nucleoproteins. [41,42]. piRNAs are transcribed into long transcripts by RNA polymerase II and then exported to the cytoplasm and cut into smaller pieces (mature piRNAs) in an unknown but most likely Dicer-independent manner [15,43,44]. piRNAs received their name due to their interaction with P-element-induced wimpy testis (PIWI) proteins, a germ-line-specific Argonaute family of nucleoproteins. [41,42]. piRNAs form RNA–protein complexes with the piwi-subfamily of Argonaute proteins. In contrast to the ubiquitous Ago subfamily, the piwi subfamily of Argonaute was found in animals. Two isoforms of PIWI proteins were described—PIWI1 and PIWI2. The overexpression of PIWI1 is associated with cell cycle arrest and the overexpression of PIWI2 is associated with anti-apoptotic signaling and cell proliferation. piRNA plays a role in RNA silencing via the formation of an RNA-induced silencing complex (RISC). Precursors are processed into mature piRNAs by two presumed mechanisms: (i) primary synthesis and (ii) ping pong amplification. For the maturation of piRNA, certain mitochondrial proteins are essential, namely Zuc, Mino, GasZ, and Armi. The piRNA molecule is 5′ monophosphated and 2′-O-methyl modified in the 3′ terminal, processes which have been proposed to increase piRNA stability. Mature piRNAs in cytoplasm form a complex with PIWI proteins and migrate back into the nucleus.

***Function.*** The accumulated data predict the role of piRNA in: (a) *genome rearrangements*, (b) *epigenetic regulation*, (c) *protein regulation*, (d) *transposon silencing*, and (e) *the maintenance of spermatogenesis* [43]. piRNAs are likely involved in the *gene silencing (of transposons)* [41,42] piRNAs have been shown to be implicated in the silencing of retrotransposons, at both the post-transcriptional and epigenetic levels, as well as the silencing of other genetic elements in germ lines, particularly during spermatogenesis. Most piRNAs show antisensing qualities in transposon sequences and are targets of the piRNAs, mainly during embryogenesis. Transposable elements risk damaging the genome through their transposition; therefore, piRNAs are essential for the protection of genome integrity. piRNAs can be transmitted into maternal cells and may be involved in maternally derived epigenetic effects [43,45].

***Clinical applications.*** The research data indicated that the overexpression of specific piRNAs can be associated with particular pathological states and cancers. piRNAs are associated with cancer development [15,45]; in breast cancer, piR-36249, piR-34736, piR-35407, piR-36318, piR-34377, piR-36249piR-36743, piR-36026 and piR-31106 were found to be significantly differentially expressed between tumours and matched non-malignant tissue. piR-651 and piR-823 were found to be overexpressed in gastric cancer paired tissue, colon, lung, and breast cancer tissues, as well as hepatic carcinoma, mesothelioma, cervical, breast, and lung cancer cell lines. piR-32051, piR-39894, and piR-43607 belong to the same cluster on chromosome 17, which is up-regulated in kidney cancer [15,43,45].

### 2.5. Small Nuclear RNAs

*Short nuclear RNA* (SnRNA) or *spliceosomal RNA* is group of single-stranded ncRNAs with an average size of 150 nt, which plays pivotal roles in the assembly of spliceosomes (RNA–RNA and RNA–protein interactions). These ribonucleoprotein complexes perform the precise removal of introns and join various exons of polyexonic pre- mRNA transcripts of the protein-coding genes [46,47,48]. snRNAs are mainly responsible for the correct positioning of the spliceosome on substrate pre-mRNAs. SnRNAs are formed from intronic parts of primary single-stranded pre-mRNA transcripts (catalyzed by RNA polymerase II) (Figure 2). SnRNAs differ from protein-coding gene mRNA transcripts in that the transcript of the snRNA gene is not spliced and the 3‘-terminal is not phosphorylated, which may be to prevent translation. Future snRNAs obtain their specific secondary structure and are embedded into complexes with specific nucleoproteins (small nuclear ribonucleoproteins (snRNPs)). A mature snRNA–snRNP complex is released from the pre-RNA molecule and starts its role in spliceosomes. The alternative splicing of the pre-mRNA of protein-coding genes is an evolutionary stem to produce multiple protein variants (many dozens from one gene). Based on shared sequence characteristics, protein cofactors, and functions, snRNAs can be divided into *Sm class* (including U1, U2, U4, U4atac, U5, U7, U11, U12) and *Lsm class* (including U6, U6ata). Based on the frequency of their use, there are major spliceosomal snRNAs (U1, U2, U4, U5 and U6), and two of the four minor spliceosomal snRNAs (U11, U12, U4atac and U6atac) [49]. U4atac ncRNA, often forming a base-paired complex with U6atac ncRNA, is a major component of U12-type spliceosome complex (much lower in abundance: ~10,000 copies per cell) required for the removal of the rarer class of eukaryotic introns (AT-AC, U12-type) [47,48,49].

***Function.*** The number of various protein isoforms is at least 5–10-fold higher than number of proteins encoded by the human genome. More than 90% of human protein-coding genes produce multiple mRNA isoforms, and this evolutionary move heavily implies alterative splicing. snRNA is a principal *part of the alternative splicing machinery*. SnRNAs are highly expressed in humans during proliferation (cell cycle) and cell development, participating in the regulation of cell fate decisions, and usually exist in clusters in the genome [46].

***Clinical application.*** The dysregulation of spliceosome components including spliceosomal ncRNAs can lead to a variety of pathologies [50,51]. Missplicing, as evidenced by intron retention, occurs in mammals in nearly 75% of multi-exon genes. Defects in *cis-acting splicing* may lead to β+-thalassemia, variants of degenerative muscle dystrophies (limb girdle muscular dystrophy), Duchenne- or Becker-type muscular dystrophies, cystic fibrosis, frontotemporal dementia with parkinsonism, or familial dysautonomia dilated cardiomyopathy (DCM). Defects in *trans-acting splicing* may lead to spinal muscular atrophy or amyotrophic lateral sclerosis (ALS) [49]. Mutations in the U4atac snRNA and defective U12 splicing can cause *microcephalic osteodysplastic primordial dwarfism type I* (*MOPD I*), also called Taybi–Linder syndrome (TALS), with developmental brain and skeletal abnormalities. The dysregulation of alternative splicing is an important factor in several types of cancers, affecting either *tumourignesis* (growth of primary tumour) or progression (local spread) and *metastasis* (e.g., PRPF6, a U5 snRNP protein overexpressed in colorectal carcinoma, and PRPF8 somatic mutations in myelodysplastic syndromes) [51]. Interestingly, 5- Fluorouracil, used in the treatment of a variety of solid tumours such as colorectal, breast, and liver carcinomas for the past six decades, may effect RNA processing and spliceosome mechanism [52].

### 2.6. Small Nucleolar RNAs

*Small nucleolar RNAs* (*snoRNAs*) are a class of highly conserved, stable, non-coding RNA molecules (60–300 nucleotides in length) that are located in the nucleolus and very abundant in many organisms [53]. They were among the first identified small ncRNA molecules. snoRNAs are generally transcribed by *RNA polymerase II* from dsDNA. SnoRNAs are either intronic or intergenic; most of the snoRNAs in humans are intronic. In human cells, most snoRNAs are excised from the introns of structural protein-coding genes, which are spliced with debranching and exonucleolytic processing, and the lariat is further processed to a mature snoRNP [53,54].

Each snoRNA associates with at least four core proteins in an RNA/protein complex referred to as a *small nucleolar ribonucleoprotein particle* (*snoRNP*). snoRNA are heavily involved as guide in the sequential chemical modification of various RNAs. Therefore, every snoRNA molecule has to precisely recognize its target and contains an antisense element (a stretch of 10–20 nucleotides), which is complementary to the sequence in targeted RNA [54]. This enables the snoRNP to recognize and bind to exact physical location in the target RNA. There are two main classes of snoRNA, the *C/D box snoRNAs* (genes encoded in the 15q11–q13 region), which are mostly associated with methylation, and the *H*/*ACA box snoRNAs*, which are associated with the pseudouridylation of target molecules [54,55,56].

***Function.*** snoRNAs are presumed to play several roles: (a) snoRNAs are involved in synthesis and guide the posttranscriptional chemical modification of other *small ncRNA molecules*, such as ribosomal RNAs (rRNA), transfer RNAs (tRNA), and small nuclear RNAs (snRNA) [54]. For example, nascent pre-rRNA molecules undergo, after transcription, a series of processing steps to produce the mature rRNA molecule. Prior to cleavage by exo- and endonucleases, the pre-rRNA molecule undergoes a series of nucleoside modifications by methylations and pseudouridylations that is guided by snoRNAs [57]. snoRNAs may serve as *miRNAs precursors*, e.g., snoRNA called ACA45 can be processed into a mature miRNA of 21 nt length by the RNAse III endoribonuclease dicer. snoRNAs are also involved in the *regulation of alternative splicing* of the trans-gene transcript, e.g., snoRNA named HBII-52, known as SNORD115), and potentially other functions.

***Clinical applications.*** Historically, it was Gallagher et al., in 2002, who proposed that the deletion of paternal allele SNORD116/PWCR1/HBII-85 could lead to Prader–Willi syndrome (PWS) as a later canonical example of the epigenetic parental imprinting mechanisms [58,59]. snoRNAs are characterized by their stability in body fluids and their clinical relevance, and represent promising tools as diagnostic and prognostic biomarkers. Alterations in snoRNA expression can affect numerous cellular processes, including cell proliferation, angiogenesis, fibrosis, and inflammation, making them a promising target for the diagnostics and treatment of various human pathologies. In recent years, data have suggested that the dysfunction of snoRNAs plays a pivotal role in tumourigenesis and related disorders. snoRNAs have been widely implicated in the pathogenesis of *hepatocellular carcinoma* (*HCC*) and related liver disorders, such as viral hepatitis B and C and non-alcoholic fatty liver disease (NAFLD) [59]. Growing evidence suggests that snoRNAs act as oncogenes or tumour suppressors in hepatocellular carcinoma (HCC) [59]. SNORD126 and SNORD105 are significantly up-regulated in HCC tissue, whereas SNORA52, SNORD31 and SNORD113 are significantly down-regulated [60,61]. Overexpressed SNORD12B acquired the largest significance as a “driver” role *in colorectal carcinoma*. The high expression of SNORA42 is related to the distant metastasis and poor prognosis of CRC. The overexpression of SNORD44, SNORA15, SNORA41, and SNORD33 may be involved in the progression from chronic intestinal inflammation to colorectal carcinoma. The expression of SNORA71A was significantly correlated with the TNM stage and lymph node metastasis [61,62]. SNORD15A, SNORD15B, SNORD22, SNORD17, and SNORD87 are overexpressed in *human breast cancer* as well as prostate cancer. SNORA71A and 71B are associated with metastatic breast cancer. SNORD3A and SNORD118 regulate tumour occurrence through the inhibition of p53 TSG. In contrast, the expression of SNORD46 and SNORD89 is significantly decreased in breast cancer tissues. SNORD50A delayed the proliferation of breast cancer and improved prognosis [59,63]. SNORD34, SNORD35A, SNORD43, SNORD49A, SNORD55, SNORA74A, SNORD105, SNORD104, SNORD110, and SNORD116-18 show various degrees of association with T-ALL, B-ALL, and CLL leukemias. SNORD42 is a diagnostic marker of AML. SNORD112-114 is mainly associated with leukemias based on stem cell pluripotency [64,65,66,67]. The expression of SNORD33, SNORD66, and SNORD76 in the plasma is significantly enhanced in non-small cell lung cancer (NSCLC) compared with that in healthy individuals. SNORA47, SNORA68, and SNORA78 are used to predict the overall survival rate of NSCLC. SNORA42 has been proven to be an oncogene in lung cancer. The expression of SNORA65, SNORA7A, and SNORA7B is also increased in NSCLC [60,68]. The expression of SNORD42, SNORD44, SNORA55, SNORD78, SNORD74, and SNORD81 is significantly increased in *prostate cancer* [69]. *In glioma tumours*, overexpressed SNORD47 may serve as an oncogene negatively correlated with the stage of glioma and positively correlated with survival time, while SNORD44 may be protective and significantly down-regulated in glioma. This is similar to the effect of SNORD 76 [68,70].

### 2.7. Long Non-Coding RNA

*Long non-coding RNA* (*lncRNA*) is a class of single-stranded ncRNAs that are typically longer than 200 nucleotides and do not encode protein. A long-mysterious group of RNAs, lncRNAs are an abundant group of ncRNAs, counting ~60,000 in the human body [71]. Many have been identified in normal tissues and cancer cell lines, yet a significant majority (more than 70%) is still awaiting appropriate analysis [72]. LncRNAs are supposed to play a crucial role in vital cell activities, such as: (a) cell cycle, cell division and proliferation, (b) regulation of cell differentiation (stem cell pluripotency), and (c) cell metabolism [71,73,74].

LncRNAsare coded from *intergenic dsDNA* (*LincRNAs*), from *intronic regions* of protein-coding genes, in an antisense manner (asLncRNAs), or in a bidirectional way (bLncRNAs) (Figure 3). Transcripts are often similar to mRNAs as they are transcribed by RNA *polymerase II*, capped and polyadenylated. The molecule contains typical splicing sites and introns/exons, and are also of a similar length to mRNA [73,75]; however, their expression level is usually lower than that of mRNA. They were originally thought to be unstable, but their structure is stabilized by polyadenylation and/or by a secondary structure [12,74]. Enhancer RNAs (eRNAs), first detected in the 2010s, represent a subclass of Lnc RNA (50–2000 nt in lenght) transcribed from the DNA sequences of enhancer regions. eRNAs can be subdivided into two main classes, 1D eRNAs and 2D eRNAs, which differ in their size, transcriptional directionality and polyadenylation place. eRNAs play active roles in transcriptional regulation (positioning of transcription factors) [71].

***Function.*** LncRNAs operate in both the nucleus and cytoplasm, being able to interact with DNA, RNA, and proteins. They are involved in the regulation of gene expression at both the transcriptional and epigenetic levels. Additionally, gene-regulatory elements are embedded in lncRNA genes [71]. LncRNAs can also act post-transcriptionally by acting as sponges for different miRNAs, such as let-7 [76]. Finally, lncRNAs can regulate the translation and posttranslational modification of proteins in the cytoplasm [74].

***Clinical application.*** Many lncRNAs exhibit tissue-specific expressions, including cancers, making them potential diagnostic and/or therapeutic targets. LncRNAs were shown to enhance cancer development and survival by promoting proliferation, growth, migration and drug resistance. LncRNAs can act as tumour suppressors (TSG) or enhance other TSGs; their activity can be downregulated in cancer (e.g., GAS5 in breast cancer and glioblastoma) [12]. Telomeric Repeat-containing RNAs (TERRAs) negatively regulate telomerase [72]. Maternally Expressed 3 (MEG3) is a TSG that can be corroborated with p53 and is downregulated in several cancers [12,77,78]. NAMA and PTCSC3 are dysregulated in thyroid cancer; lncRNA-p21 is dysregulated in rectal cancer [12]. The lncRNA ARHGAP27P1 inhibits gastric cancer cell proliferation and cell cycle progression through the epigenetic regulation of p15 and p16 [75]. 

LncRNAs work also as inhibitors of TSG, e.g., HOTAIR and retinoblastoma (RB) [12]. The overexpression of lncRNA HOTAIR can promote the development of gastric cancer through the inhibition of TSGs, such as the progesterone receptor, protocadherin 10, protocadherin β5 and junctional adhesion molecule 2 [79]. LncRNA MALAT1 can promote gastric cancer metastasis by inhibiting PCDH10 [12]. LncRNA FAM230B promotes gastric cancer growth and metastasis by regulating the miR-27a-5p/TOP2A axis [12,76]. 

LncRNAs serve as oncogenes (ONC), e.g., LUCAT1 is highly expressed in various malignant tumours (breast cancer, ovarian cancer, thyroid cancer, and kidney cancer) [80]. LUNAR1 increases the action of IGF1 in T-cell acute lymphoblastic leukemia [76]. NEAT1 is a target of p53 and is overexpressed in liver and ovarian cancer and melanoma [76]. LncRNAs may also enhance the effects of ONC products, e.g., PVT1 prolongs the viability of MYC oncogene [12,76]. 

LncRNAs may promote cancer by interfering with miRNA, e.g., the increase in miR-675 caused by H19 in hepatocellular carcinoma and rectal cancer. The dysregulation of lncRNA can affect the treatment of cancers by drugs, e.g., through the silencing of PTEN, lncRNA PCAT-1 promotes the cisplatin-resistance of gastric cancer [12]. Through the inhibition of miR-217, lncRNA HOTAIR increases the resistance of gastric cancer cells to doxorubicin and paclitaxe [79].

## 3. ncRNA Applications in Particular Clinical Areas

### 3.1. Oncological Diseases

Non-coding RNAs have been extensively studied in the context of various cancers (Figure 4) based on their involvement in tissue development [81], cell proliferation, differentiation [82], and apoptosis [83]. The expression of miRNAs, their amount, and their type, when found in healthy cells and tissues, differs significantly from those found in tumours. Different types of tumours are characterized by an altered representation of miRNAs, which can be studied as tumour biomarkers [84].

Current research in this topic is always exploring new microRNAs as potential biomarkers. A combination of four microRNAs—miR-1246, miR-206, miR-24, and miR-373,—has been reported to detect early-stage breast cancer with 98% sensitivity, 96% specificity, and 97% accuracy [85]. An increased expression of miR-27a, miR-206, and miR-214 has been associated with enhanced tumour chemoresistance and shortened patient survival in patients with triple-negative breast cancer [86]. Non-coding RNAs may also serve as monitoring tools in the treatment and management of oncological diseases, surveilling early responses to neoadjuvant chemotherapy from blood and urine samples. piR-36743, miR-17, miR-19b, and miR-30b have the potential to act as prognostic biomarkers in patients with triple-negative breast cancer [87]. Piwi-interacting RNAs can potentially act as therapeutic agents for breast cancer. piR651 has been shown to be dysregulated in various cancer tissues, including gastric and lung cancer. It appears that, in breast cancer, piR-651 has a dual character. Overexpression leads to tumour growth and metastasis, caused by the upregulation of MDM2, CDK4 and Cyclin D1. However, piR-651 appears to promote PTEN methylation and its downregulation by recruiting DNMT1. This induces the apoptosis of breast cancer cells, acting as a potential therapeutic agent for breast cancer [88]. piRNA-36712 is a tumour-suppressor that is down-regulated in breast cancer cells, targeting SEPW1. The higher expression of SEPW1 due to the down-regulation of piRNA-36712 promotes cell proliferation, invasion, and migration via upregulated Slug, decreased p21 and p53, and E-cadherin levels [89]. In recent years, long non-coding RNAs have been studied as possible markers for breast cancer metastasis, including HOTAIR [90], NR2F1-AS1 [91], ZEB2-AS1 [92], CCAT2 [93], H19 [94], and BLACAT1 [95].

***Lung cancer*** is the most frequent malignancy and a leading cause of cancer death in the male population worldwide. Early stages of lung cancer are often asymptomatic and can only be detected by imaging methods. A microRNA diagnostic assay consisting of miR-1268b and miR-6075 is showing great potential in an early non-invasive screening of lung cancer, with a sensitivity and specificity of 99% regardless of histological type or stage of the tumour [96]. Piwi-interacting RNAs play a role in lung carcinogenesis; however, their involvement has not been fully defined. Significantly increased levels of piR651 have been observed in non-small cell lung cancer [97]; its inhibition has resulted in a decrease in the proliferation and migration of tumour cells [98], making it a potential biomarker and therapeutic agent of lung cancer [99]. piRNA-14633 is a promising prognostic marker in patients with non-small cell lung cancer, as its expression is significantly lowered in tumour cells compared to normal tissues. Down-regulation has correlated with the clinical stage and histological grade of tumour, along with lymph node invasion, increasing the severity of the disease [100].

The expression of small nuclear RNAs U1, U2 and U5 is down-regulated in patients with lung cancer. Expression levels of snRNAs isolated from patients’ plasma and healthy controls were quantitated and compared in a study by Dong et al. This dysregulation is correlated with lung cancer progression, serving as a promising biomarker [101]. Fragments of snRNA U2 can serve as lung cancer biomarkers, considering their high relative expression serum levels in lung cancer patients, with a sensitivity of 86% and a specificity of 81% [102].

***Colorectal cancer*** is the third most frequent cancer subtype and fourth most common cause of cancer death in both sexes worldwide. Over 50% of patients with colorectal cancer die due to late diagnosis; diagnosis can be made either by obtaining a sample of the colon, screening stool samples, or using imaging methods to determine the spread of the disease. Non-coding RNAs may serve as diagnostic or prognostic tools in colorectal cancer. Levels of miR-92a and miR-144 are significantly dysregulated in stool samples of patients with colorectal carcinoma compared to healthy controls, with a diagnostic sensitivity of 89.7% and 78.6%, and a specificity of 51.7% and 66.7%, respectively [103]. Aberrantly expressed piRNAs have been connected to the hallmarks of cancer phenotype and carcinogenic process. PiRNA-823 has been previously reported to be dysregulated in hepatocellular carcinoma and multiple myeloma, promoting tumour growth and metastasis. A recent study has shown that the inhibition of piRNA-823 leads to the inhibition of proliferation, invasion and apoptosis in colorectal cancer. On the other hand, its overexpression correlated with poor overall survival and predicted poor response to adjuvant chemotherapy in patients with colorectal cancer [104]. Based on their aberrant overexpression, several piRNAs have been dubbed as potential early diagnostic markers for colorectal cancer from serum or tissues, piR-24000 [105], piR-020619, piR-020450 [106], piR-54265 [107]. A piwi-interacting RNA expression assay from serum samples consisting of piR-020619 and piR-020450 appears to be capable of detecting early-stage, small-sized tumours, making them another candidate for an early diagnostic tool [106]. Low levels of certain long non-coding RNAs have been associated with poor patient outcome [108], with their involvement in tumour progression making them not only a diagnostic biomarker, but also a therapeutic target [109,110].

### 3.2. Neurodegenerative Diseases

A major fraction of human miRNAs are specifically expressed in the nervous tissue; more than 70% of all miRNAs are expressed in the brain, making these molecules potential biomarkers for the differential diagnosis and prognosis of several neurodegenerative diseases and brain/spine injuries [111]. The differences in the localization-based expression and dynamic regulation of miRNAs during brain development indicates their essential roles in neuronal development and various functions, such as synaptic plasticity, learning and memory [112]. miR-425-5p and miR-502 have been shown to be significantly downregulated in serums of patients with traumatic brain injury during early stages, making them ideal potential markers for mild traumatic brain injury. miR-21 and miR-335 have been shown to be significantly upregulated in serums of patients with traumatic brain injury, making them potential markers of severe traumatic brain injury [113]. Significant upregulation has also been observed in miR-103a-3p, miR-219a-5p, miR-302d-3p, miR-422a, miR-518f-3p, miR-520d-3p, and miR-627 in serums of patients with traumatic brain injury compared with their expression in controls. Particularly significant among these seven microRNAs is miR-219a-5p, which could discriminate patients from controls and also distinguish the severity of the injury (severe or mild) [114].

Neurodegenerative diseases are caused by progressive neuronal damage, resulting in loss of the structure or function of neurons, and ultimately leading to cell death [115]. Neurodegeneration is a complex process, involving several components, including different classes of non-coding RNAs. Mutations of small nuclear RNA U2 cause neuron degeneration as a result of flawed pre-mRNA splicing, which is normally regulated by U2 [116]. Spinal muscular atrophy is caused by the neurodegeneration of spinal motor neurons, a result of SMN1 gene mutation.s The SMN protein is responsible for the assembly of sm-class snRNAs, causing splicing dysfunctions. The SMN1 gene and its product, the SMN protein, are potential therapeutic targets for spinal muscular atrophy, and the dysregulation of snRNAs may serve as a potential biomarker [117].

Alzheimer’s disease is the most common neurodegenerative disease in older adults, characterized by a loss of neurons and synapses in the cerebral cortex. Pathological alterations take place years before the appearance of clinical symptoms; early diagnosis and treatment have not been proven successful to date [118]. Studies of microRNA dysregulation in patients with Alzheimer’s disease showed a significant deregulation of miR-30b-5p, miR-22-3p, and miR-378a-3p [119], decreased expression of miR-103 in Alzheimer’s disease patients in comparison to patients with Parkinson’s disease and healthy controls, decreased expression of miR-107 in patients with Alzheimer’s disease as well as Parkinson’s disease [120], and decreased expression of miR-223-3p in comparison with its upregulation in patients with Parkinson’s disease [121]. Jain et al. succeeded in defining a combined microRNA and piwiRNA expression signature from human cerebrospinal fluid that could be used to diagnose patients with Alzheimer’s disease. This signature includes miR-27a-3p, miR-30a-5p, miR-34c and piR_019324, piR_019949, and piR_020364 [122]. Possible roles of piwi-interacting RNAs are still emerging; RNA sequencing has revealed the five most upregulated piRNAs (piR-61646, piR-31038, piR-33880, piR-34443, and piR-37213) in patients with Alzheimer’s disease, with the potential to act as future diagnostic biomarkers [123]. In a recent study, Cheng et al. report increased levels of long-noncoding RNAs in the serums of patients with Alzheimer’s disease, namely ASMTL-AS1, AP001363.1, AC005730.3, and AL133415.1, and consider them potential biomarkers in the Chinese population [124].

Parkinson’s disease is a chronic neurodegenerative disorder of the central nervous system, affecting both motor and non-motor systems, and the second most common neurodegenerative disorder. The disease is caused by the selective death of dopaminergic neurons in substantia nigra in the midbrain region [125]. miR-132-3p and miR-146a-5p have been shown to be promising potential biomarkers for the early diagnosis of Parkinson’s disease, because of their significant down-regulation in serums of patients with severe Parkinson’s disease [126]. Yan et al. has previously reported seven deregulated microRNAs in serums of patients with Parkinson’s disease (miR-30c, miR-31, miR-141, miR-146b-5p, miR-181c, miR-214, and miR-193a-3p) and recently, they furthered their research and compared their expression in patients with Parkinson’s and multiple system atrophy, revealing a lower expression of miR-31, miR-141, miR-181c, miR-193a-3p, and miR-214 in multiple system atrophy than in Parkinson’s disease, accurately discriminating between them [127]. A study by Boros et al. has shown significantly increased levels of long non-coding RNA NEAT1 (core element of nuclear paraspeckles) in the peripheral blood of patients with Parkinson’s disease. The aim of the study was to determine a biomarker that is easily detectable by routine laboratory testing [128]. The involvement of long non-coding RNAs is hypothesized to affect neuronal degeneration by promoting apoptosis of dopaminergic neurons [129].

### 3.3. Cardiovascular Diseases

Cardiovascular diseases (CVDs) are one of the leading causes of death globally. Therefore, it is essential to identify and characterize promising new avenues to better investigate the emergence and progression of CVDs.

The crucial roles of lncRNAs in heart development, pathophysiology, and lipid metabolism reveal their promise as novel diagnostic biomarkers of disease. Abnormalities in lncRNAs’ function and expression are associated with cardiovascular biology and disease [130]. The role of lncRNAs in cardiovascular disease is becoming apparent; there is particular interest in the dysregulation of lncRNA regulatory circuits in cell fate, cardiac hypertrophy, vascular disease, atherosclerosis, and metabolic syndrome [130,131,132].

lncRNAs have been investigated in the context of cerebrovascular diseases. In one of the first studies of lncRNA expression from whole blood samples in ischemic stroke (compared to controls), a significant number of lncRNAs were detected to be differentially expressed according to post-stroke status, with some differentially expressed lncRNAs situated in close genomic proximity to putative stroke risk genes [133,134]. An altered lncRNA expression was found in brain microvasculature under stroke conditions in an in vitro model of oxygen–glucose deprivation in primary brain microvascular endothelial cells [135]. The most upregulated lncRNAs include small nucleolar RNA host gene 12 (Snhg12), Malat1, and lnc-oxygen–glucose deprivation (lnc-OGD) 1006, while the most downregulated lncRNAs include 281008D09Rik, paternally expressed 13 (Peg13), and lnc-OGD 3916, a fact that was confirmed in cerebral micro vessels isolated from mice after experimental stroke [130].

*Mhrt* is a cardiac-specific lncRNA expressed in mouse and human adult cardiac tissue. This type of lncRNA participates in regulating genes such as *Myh6*, *Myh7*, and osteopontin (SPP1 or OPN) through its effects on the chromatin remodeling factor Brg1. *Mhrt* is silenced during pressure-overload-induced hypertrophy, and restoring its expression is cardioprotective in mice. Therefore, *Mhrt* levels may be a potential biomarker to predict heart failure in humans [130,133].

lncRNAs could also play a role in the pathogenesis of ischemic stroke. As an example, the lncRNA Malat1 may play a protective role in ischemic stroke through the inhibition of proapoptotic proteins and proinflammatory cytokines, suggesting that lncRNAs may offer novel targets to reduce stroke-related brain damage [135]. The main molecular mechanisms of lncRNAs in the regulation of ischemic brain injury include the epigenetic regulation of transcription via chromatin, influencing miRNA expression, and the translational repression of certain target mRNAs [136,137,138].

Recently, piwi-intracting RNAs have been found to be differentially expressed in cardiovascular diseases. In patients with thromboembolic pulmonary hypertension (CTEPH), an increased expression level of DQ593039 has been observed in extracellular vesicles. This observation correlated with clinically significant parameters of CTEPH; therefore, DQ593039 may act as a potential diagnostic marker for pulmonary hypertension and disorders of the right heart [139]. Human piRNAs piR-020009 and piR-006426, isolated from serum exosomes of patients with heart failure, are significantly downregulated compared to healthy controls, possibly acting as therapeutic agents in the suppression of heart failure progression [140]. A study by Rajan et al. examined the expression of piRNAs isolated from serum in both in vitro and in vivo conditions in patients with cardiac hypertrophy. Their results extended their previous research, providing insight into piRNAs’ role in myocardial infarction. PiRNAs are associated with AKT and Stat3/bcl-xl signaling pathways and may regulate ischemic heart diseases via acting upon downstream targets of these pathways [141].

### 3.4. Diabetes Mellitus

Type 2 diabetes mellitus (DM2) is a manifestation of insulin resistance, which demonstrates itself through a faulty response to insulin and subsequent altered glucose metabolism. It is a consequence of obesity, metabolic syndrome and altered lipid homeostasis. The metabolic pathway of insulin resistance leads from insulin to glucose transporter type 4 (GLUT4), which regulates glucose uptake. Each of these steps is regulated by several miRNAs that play an important role in pathogenesis and are a prerequisite for their use in therapy [142].

Patients with DM2 exhibit abnormalities in tissue sensitivity to insulin, as well as insulin secretion, by the pancreas, as pancreatic beta cells adapt to changes in insulin action (decreased insulin action means increased insulin secretion and vice versa). Any alterations in this dynamic interaction lead to dysfunctional glucose metabolism [143]. Studies investigating glucose metabolism in DM2 have examined the expression profiles of several miRNAs implicated in the pathogenesis of DM2, comparing newly diagnosed DM2 patients, pre-DM2 subjects, and DM2-prone subjects with normal glucose tolerance. The authors noted a significantly higher expression of miR-9, miR-29a, miR-30d, miR-34a, miR-124a, miR146a, and miR-375 in DM2 patients compared to DM2-prone subjects. In addition, some miRNAs were significantly down-regulated in pre-DM2 compared to DM2 patients. They also found that the expression patterns of circulating miRNAs in patients in the pre-DM2 group were similar to those observed in patients with normal glucose tolerance. On the other hand, the expression levels of the mentioned miRNAs do not change dramatically in the pre-DM2 stage, which reduces their usefulness as a DM2-specific biomarker [144,145,146]. In another study, the authors aimed to identify patients with metabolic syndrome compared to patients with DM2. miR-150, miR-192, miR-27a, miR-320a, and miR-375 were up-regulated in DM2; thus, they have the potential to participate in the regulation of hyperglycemia [146]. Another study looked at the miRNA profile in normoglycemic, DM2-prone, and DM2 patients. Although the expression of several circulating miRNAs was investigated, only three miRNAs, miR-15a, miR-223, and miR-126, were detected in blood plasma. Levels of miR-126 were analogous between the pre-DM2 and DM2 groups, and significantly lower than in the normoglycemic group. At the same time, the expression pattern of plasma miR-126 was associated with fasting glucose level. The authors concluded that miR-126 could be a potential non-invasive diagnostic tool for the prediction and prevention of DM2 [147,148].

In their study, Liu et al. (2014), in addition to monitoring the differential expression of miR-126 as a potential biomarker for pre-DM2 and DM2, also evaluated the possible therapeutic value of miR-126. The conclusions of their study confirm the diagnostic and therapeutic potential of miR-126 at the early stages of DM2 development [149].

According to studies, miR-375 and miR-9, which affect the regulation of insulin secretion, also show potential biomarker properties. Although, on the one hand, miR-375 is more suitable for distinguishing patients with DM2 from healthy probands, as well as patients with pre-DM2 and DM-2, it is the combination of miR-375 and miR-9 that correlates with the susceptibility to developing DM2 [150].

The presented studies confirm the diagnostic potential of miRNAs (and the therapeutic effect of mi-R126) as they are differentially expressed between persons prone to DM2, persons with DM2, and healthy individuals. Dysfunctions in the glucose metabolism have a significant impact on other cells, tissues, and processes, leading to DM2. miR-126 and miR-375 are among the most frequently monitored miRNAs; therefore, it is necessary to focus on clarifying their role in the pathogenesis of DM2.

Other non-coding RNAs are suspected to play a role in metabolism-related functions, insulin resistance and beta cell function. In a study by Barbosa et al. using a rat model (on a high-fat diet), a transcriptomic analysis of F0 fathers and their F1 female offsprings revealed changes in the expression profile of piRNAs and provided insight into transgenerational epigenetic changes affecting the metabolism of two generations via nutrition [151]. Other evidence for piRNAs’ involvement in beta cells’ function was presented via the silencing of PIWIL2 and PIWIL4 in rats, causing decreased levels of piRNAs and deficiencies in insulin secretion [152]. piRNA-48383 has been identified as a specific piRNA associated with insulin resistance in a study exploring the relationship between metabolic phenotype and circulating extracellular RNAs [153].

### 3.5. Gastrointestinal Diseases

miRNAs regulate several biological processes (cell cycle, cell proliferation, cell movement activity, biochemical processes, immunity, inflammatory processes, and apoptosis [154,155,156]). miRNAs are involved in the regulation of the gastrointestinal system (GIT) and also in its functionality; therefore, dysregulation is associated with many diseases: celiac disease, ulcerative colitis, Crohn’s disease, stomach cancer and others [157,158]. miR-143/145 and miR199a/214 clusters are involved in the regulation and proliferation of smooth muscle cells by miRNAs’ switching smooth muscle cells between proliferation and differentiation states [159].

The most frequently studied group of miRNAs in GIT diseases is the miR-29 family. miR-29a was upregulated in ulcerative colitis tissues compared with healthy tissue [160]. In a clinical study by Fasseu et al. (2010), the authors found that miRNA-29a and miRNA-29c were significantly increased in GIT tissue from patients with Crohn’s disease compared to controls [161]. IFN-γ production was also increased in patients with Crohn’s disease, and also plays an important role in ulcerative colitis via miR-29, which is upregulated in these diseases and targets IFN-γ mRNA [154]. In addition, Crohn’s disease is associated with the down/up-regulation of miR-19a, miR-1273d, miR886-5p, miR3194, miR192, and miR-200a [162]. The SOCS3 suppressor plays an important role in the inflammatory response in Crohn’s disease. miR-19b suppresses this suppressor, and thus prevents the onset and development of Crohn’s disease [163]. Schaefer et al. (2015), in their study of colon, blood, and saliva samples, suggested that miRNA may contribute to the pathogenesis of IBD or at least reflect ongoing inflammation, and that miR-19a, miR-21, miR-31, miR101, miR146a, and miR-375 could be reliable markers to identify and distinguish between Crohn’s disease and ulcerative colitis [164]. For the monitoring and diagnosis of the disease in patients with IBD or Crohn’s disease, a non-invasive examination, namely, the examination of miRNA expression in saliva and blood samples, would be sufficient. Cheng et al. (2015) conducted a clinical study on 60 IBD patients with 120 samples, where they found that miRNA-21 stimulates invasion, intravasation and metastasis in IBD-related carcinogenesis through the post-transcriptional downregulation of the tumour suppressor PDCD4 [163]. Other studies have also confirmed the role of microRNAs in colorectal cancer [165,166]. Duan et al. (2016) identified that miRNA-130 promotes cell proliferation and migration in gastric cancer [167]. Another study by Wu et al. (2015) identified that miR-421 in serum may also be a potential biomarker of gastric cancer [168].

MiR-122 is the most abundant miRNA (~70%) in hepatocytes and in the liver in general, and plays a significant role in normal metabolic functions [169,170,171]. The downregulation of miR-122 directly controls cholesterol and lipid metabolism, and miR-192, and miR-194 in hepatocytes lead to hepatocellular damage, steatosis, and liver failure [170,172]. Cirrhosis of the liver represents the common end-point of chronic liver injury, and hepatic stellate cells (HSC) play an essential role in this fibrotic process. HSC activation is associated with the upregulation of miR-29 (therapeutic inhibition) and also of miR-501, miR-349, miR-325-5p, miR-328, miR-143, and miR-193. However, others showed significant downregulation (e.g., miR-341, miR-15b, miR-16, miR-375, miR-122, miR-146a, and miR-126 [170,171,172]. MiR-103 and miR-107 are involved in hepatic insulin effects on glucose/glycogen homeostasis [170,171]. Anti-miR-103/107 (RG-125) were involved in tests for the treatment of nonalcoholic steatohepatitis (NASH) patients who suffered from type 2 diabetes [170,173].

### 3.6. Respiratory Disorders

From non-cancerous pulmonary diagnoses, the main interest in the ncRNA profiling of sputum, epithelial or immune cells, and blood specimens was expectedly recorded in common obstructive disorders such as asthma and chronic obstructive pulmonary disease (COPD) or in their overlap, as shown in Figure 5. The effect of industrial pollution and environmental irritants and allergens was also the focus of ncRNA analysis [174]. These chronic life-time disorders belong to those with high morbidity in young and adult population and in the case of COPD, which also shows the highest mortality worldwide.

In asthma, Th2, an allergic disease, routinely increased serum levels of miR-21 (switch of Th1 vs. Th2), and miR-155 miR-155 (Th2 cell maturation, type 2 innate lymphoid cells ILC2s, eosinophiles) were found—important regulators of the expression of many immunologically relevant genes. Upregulated miRNAs also include the miR-223 family (inflammation machinery), miR-146 family (genetic endotype of asthma), miR-142-5p, miR-142-3p, miR-629-3p (severe asthma), miR-221, and miR-143-3p (inhibit airway remodeling) [174,175,176]. Downregulated miRNAs included the let-7 family, miR-193b, miR-375, or miR-192-5p [174]. lncRNA CASC2 and BAZ2B are mostly increased in the serum of patients with childhood asthma [177,178]. COPD patients display upregulated miR-497-5p, miR-130b-5p, miR-126-5p miR-221-3p, miR-92a-3p, miR-145, miR-320c, miR-200c-3p, and miR-449c-5p. However, miR-15b-5p appeared to be most consistently downregulated [174,179]. Wang et al. reported that, from 148 significantly identified miRNAs in the blood of COPD patients, 104 miRNAs were upregulated, and 44 miRNAs were downregulated [180]. Asthma and COPD overlap syndrome often reveals upregulated miRNA-338 in the sputum, while the expression of some other miRNAs is lowered (miR-148a-3p, miR-15b-5p, miR-223-3p, and miR-23a-3p) [174].

## 4. Methodologies in ncRNAs’ Detection

Currently, it is unknown how many non-coding RNAs are present in the human genome; however, new transcriptomic and bioinformatic research indicates that thousands of non-coding transcripts may exist. A large number of non-coding RNAs serve a purpose as epigenetic regulators of several cellular processes [181]. Alterations in non-coding RNAs’ expression have been shown to contribute to the dysregulation of gene expression, resulting in various human disorders [182]. Biomarkers are a measurable indicator of a normal biological or pathogenic state that provide information on responses to treatment. A suitable molecular biomarker should be stable, sufficiently specific, easily detectable, and, in particular, its use should be universally applicable to all patients [183]. Dysregulated non-coding RNAs detected in body fluids meet these criteria, and are considered potential diagnostic and prognostic biomarkers for several human diseases. The detection and quantification of ncRNAs is currently becoming easier and more efficient using methods like next-generation sequencing (NGS), RNA microarray, real-time RT-PCR, digital PCR, and Northern blot [184].

*Next-generation sequencing and RNA microarray* are capable of detecting and quantifying a series of transcripts at the same time; however, a microarray requires larger amounts of RNA and only measures known transcripts, while NGS could be used to identify novel ncRNA transcripts [185]. NGS is a sequencing technology used to determine the order of nucleotides in targeted regions of ncRNAs, using sequencing through synthesis technology, tracking the addition of labeled nucleotides as the RNA chain is copied [186]. The microarray method is used to detect the expression of genes involved in biochemical pathways. Microarrays utilize a solid substrate, which assays a certain amount of target transcripts using multiplex lab-on-a-chip method [187].

*Real-time RT-PCR* and digital PCR provide fast results and, based on their high sensitivity, require only small amounts of isolated RNAs, but can only detect and quantify a single ncRNA at a time. PCR methods amplify the target DNA obtained after the reverse transcription of sample RNA. The detection of PCR products is conducted via specific primers that are complementary to a specific sequence of DNA [188]. Shorter non-coding RNAs, with an average length of from 21 to 31 nt, have to be extended during reverse transcription by a stem loop primer or by adding a 3′poly A tail, followed by reverse transcription with a poly T primer, with a universal sequence added at the 3′end. Later, qPCR is performed with a forward primer specific to the ncRNA and reverse primer that is complementary to the stem-loop or the poly T primer [189].

*Northern blot* is currently considered a gold standard, and is mainly used for novel microRNA confirmation and quantification because of its ability to identify the size of the target RNA. However, it is a time-consuming method, requiring large amounts of RNA, and is able to detect only a single transcript at a time [190]. During Northern blot analysis, RNA samples are separated by electrophoresis according to size, transferred onto a nitrocellulose membrane, and detected with a labelled hybridization probe that is complementary to the target RNA. Upregulation or downregulation can be observed based on the abundance of RNA on the Northern blot [191].

## 5. Conclusions

Research on non-coding RNAs’ potential significance in numerous diseases has been based on their abundance in multicellular organisms, varying involvement in cellular processes, and impact on the expression of protein-coding genes. Many diseases can be brought on by changes or dysregulation in the body’s non-coding RNA reserve.

The employment of methods such as high-throughput RNA sequencing allows for the expression of mRNAs, lncRNAs, piRNAs, miRNAs, and circRNAs from the blood or other liquids to be analyzed under specified conditions to *build a map of RNA biomarkers* of specific disorders and their variants, and to distinguish in between them. Upregulated or downregulated ncRNAs increasingly appear to be reliable biomarkers for diagnostics, disease stage, therapeutical success, or prognosis. Combining them further enhances their scope, and provides more detailed disease fingerprints for differential diagnostics. As has emerged from recent developments in the field, in addition to being markers, the *multi nc-RNA* signatures help to disclose the *co-expression of circRNA*–*miRNA*–*mRNA regulatory networks* enrolled in pathogenesis-specific disease features, or as general mechanisms shared by various disorders (e.g., inflammatory machinery, immune suppression or activation, tissue reconstruction, regeneration, apoptosis, DNA repair, cell migration, proliferation, and cell-to-cell transition). In the future, such a molecular pathogenetic approach will inevitably combine the data from genomics, transcriptomics, and proteomics in a unified framework. Interpretation, however, will always require rational pathophysiological insight and clinical feedback. The biological systems, in either normal or disease conditions, are naturally variable and are permanently evolving. Distinct markers obtained from specific cells or the blood may indicate different stagings and gradings of disease, progress and/or spontaneous remissions, therapeutical regression, known or hidden comorbidities, and other epigenetic modifiers, such as environmental, lifestyle, and nutritional influences. All these provide challenging topics for further research.

## Figures and Tables

**Figure 1 ijms-24-16213-f001:**
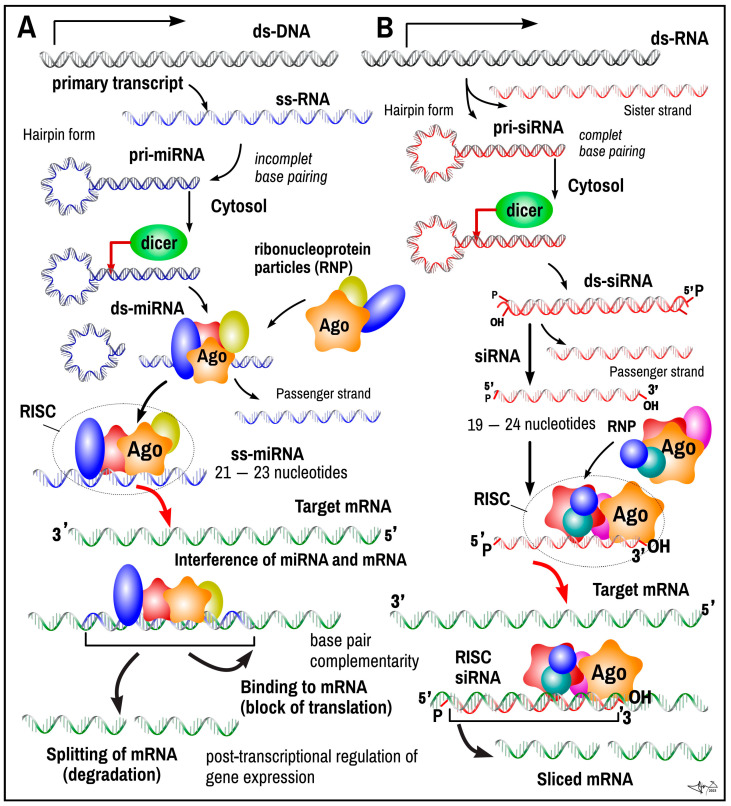
RNA interference pathway in microRNA and silencing RNA action. Primary single-strand transcript of microRNA (known as a pri-miRNA) is processed in the cell nucleus into a 70-nucleotide stem–loop structure called a pre-miRNA by the microprocessor complex (RNAse III). It is folded back to self-assemble into double-stranded form with a small loop (hairpin shape) at one end of the molecule. The complementarity of the bases (C-G; A-U) is not always perfect in miRNA, leading to several mismatched bases. siRNAs derived from long dsRNA precursors differ from miRNAs in that the base pairing is more complete. The hairpin-shaped primary miRNA (pri-miRNA) is exported out of the nucleus in the following step via exportin-5 transport. In cytosol, a hairpin loop is trimmed off by Dicer endo-ribonuclease (RNA helicase), while the rest of the double-stranded remnant of miRNA assembles into a complex with the Argonaute family of nucleoproteins. Precursors of siRNA, long pieces of ds siRNA or hair pin single-stranded siRNA, are called pri-siRNA. Dicer enzyme cuts these molecules to form short double-stranded interfering RNA or siRNA (usually from 20 to 24 bp dsRNA) with phosphorylated 5′ ends and hydroxylated 3′ ends with two overhanging nucleotides. miRNA and siRNA form complexes with nucleoproteins called RNA-Induced Silencing Complexes (RISCs). The RISC of miRNA and siRNA binds to complementary motifs in mRNA. In both cases, the posttranslational mRNA silencing process is started: (1) translation is blocked by the prolonged binding of complex RISC-miRNA, or (2) the mRNA strand is degraded by RISC-miRNA or RISC-siRNA by the splicing to pieces and/or shortening of the poly(A) tail. Legend: red—siRNA sequences; blue—miRNA sequences; green—target mRNA.

**Figure 2 ijms-24-16213-f002:**
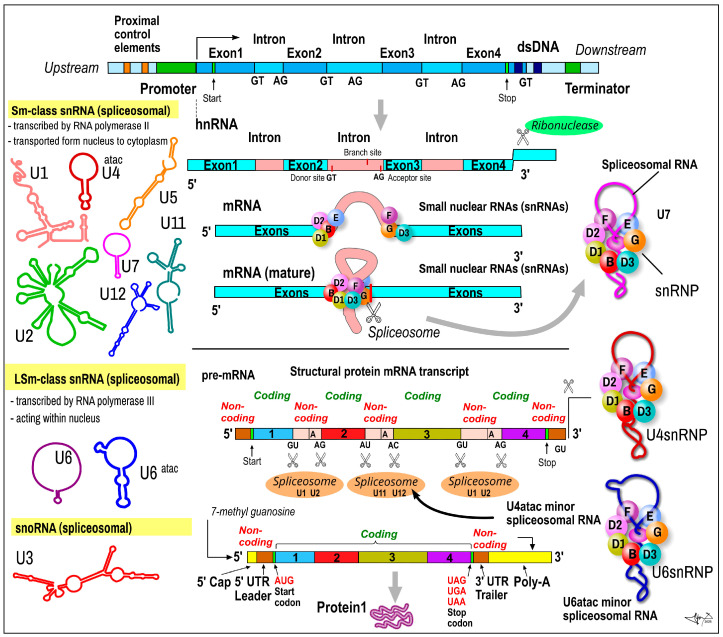
Biogenesis of small nuclear (snRNA). snRNA (average length is 150 nt) is a class of small RNA molecules that are found within the cell nucleus in eukaryotic cells. They are transcribed by either RNA polymerase II or RNA polymerase III [1]. Their primary function is in the processing of pre-messenger RNA (hnRNA) in the nucleus. Examples and classification of spliceosomal snRNA into Sm and LSm class, respectively, is shown on the left. The most common human snRNAs include U1, U2, U4, U5, and U6 spliceosomal RNAs, respectively (for mor detail, see the text). Their nomenclature derives from their high uridine content. U3 belongs to small nucleolar RNA. snRNAs are always associated with a set of specific proteins, and the complexes are referred to as *small nuclear ribonucleoproteins* (*snRNP*). snRNA participates in spliceosomes. U4atac-snRNP and U6atac-snRNP are pivotal components of U11 and U12 spliceosomes typically involved in the posttranscriptional splicing of mRNA in specific locations (AT-AC (AU-AC)).

**Figure 3 ijms-24-16213-f003:**
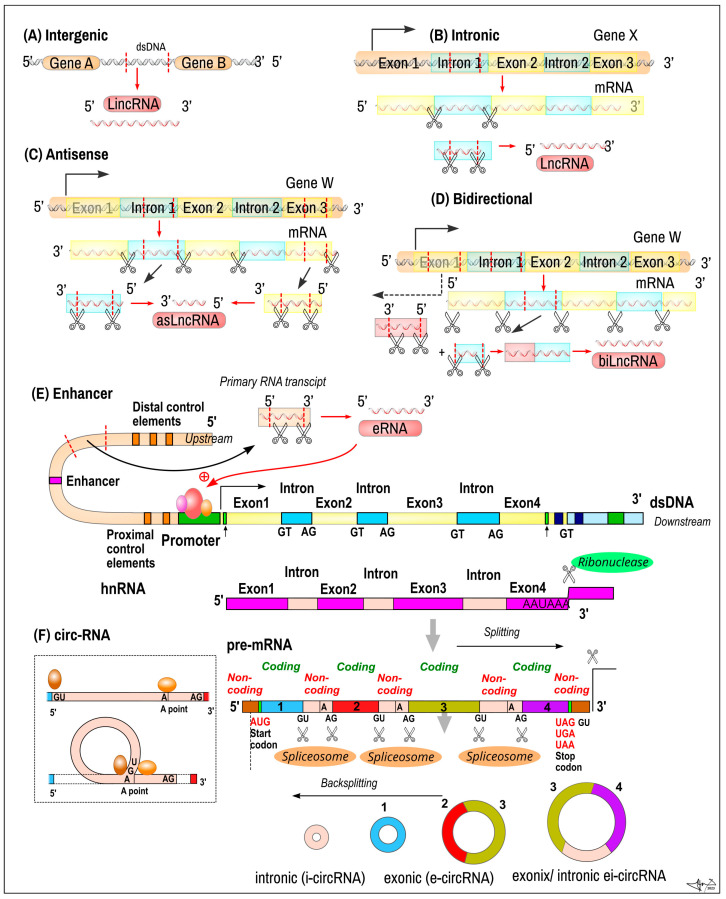
Biogenesis of long non-coding RNA (LncRNA) and circular RNA (circRNA). *LncRNA* classification depends on the genomic position: (**A**) intergenic RNAs (LincRNAs) are transcripts of dsDNA in between two protein-coding genes; (**B**) *intronic LncRNAs* are transcripts from within an intronic region of a protein-coding genes; (**C**) *antisense LncRNAs* (asLncRNAs) are transcribed from complementary strands—either from within the intronic or exonic region of protein-coding genes; (**D**) *bidirectional lncRNAs* (biLncRNA) originate from the bidirectional transcription of protein-coding genes; (**E**) *enhancer LncRNAs* (*eLncRNAs*) are LncRNAs (50–2000 nt) transcribed from dsDNA of the enhancer regions of genes. These LncRNAs mediate transcription factor positioning in promoters of protein-coding genes. *CircRNAs* are LncRNAs and can originate from primary intronic (i-circ-RNA), exonic (e-circ-RNA) or both exonic and intronic fragments (ei-circRNAs) transcripts of protein coding genes that undergo back splicing arrangement. (**F**) Formation of the circRNA loop by cutting at the 5′ end of RNA—transcript and attachment to the fusion point close to the 3′ end of the molecule. At the bottom of the figure: variants of circRNA. Numbers indicate exons. Legend: Scissors–splicing sites.

**Figure 4 ijms-24-16213-f004:**
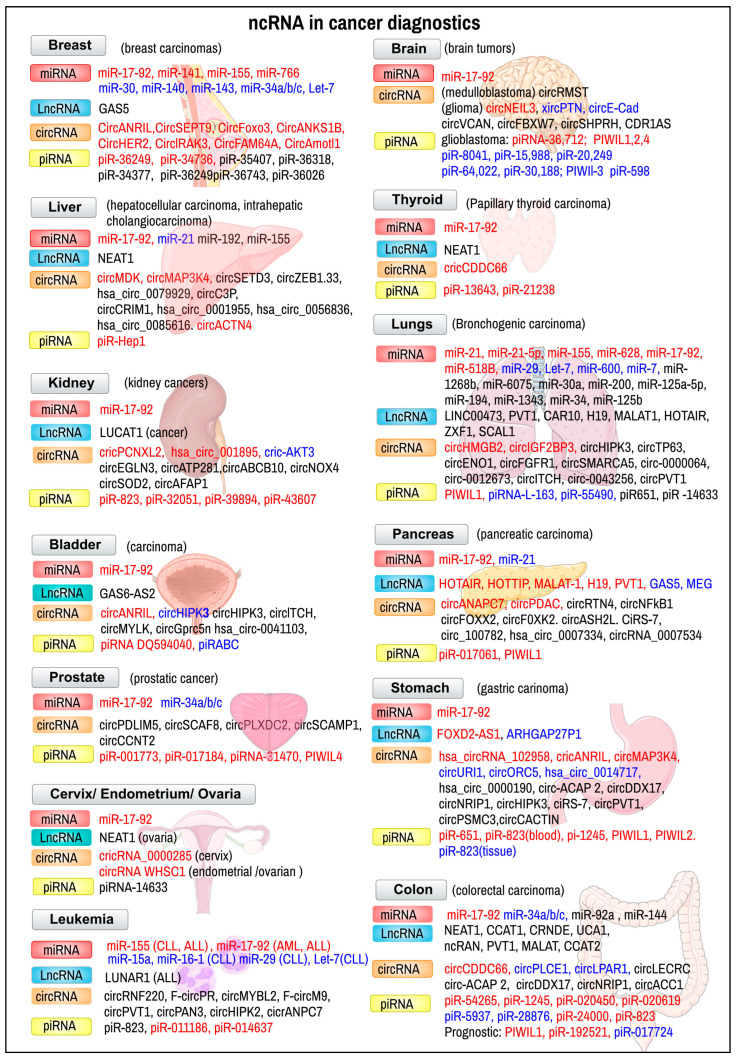
Examples of ncRNAs in cancer diagnostics. Types of ncRNAs discussed in the text are depicted. Diagnostic (or prognostic, if denoted) markers from peripheral blood, tissue (endoscopic biopsy), or other body liquids. In certain markers, their expression levels are indicated by color: red—upregulated markers associated with promoted and invasive growth (protoncogenic) or metastases; blue—downregulated markers associated with antioncogenic, tumour supressor, antimetastatic tumour activites or tumour regression. Certain pictures shown in the background are credited to Servier Medical Art, France (used and modified according to CC-BY-4.0 licence).

**Figure 5 ijms-24-16213-f005:**
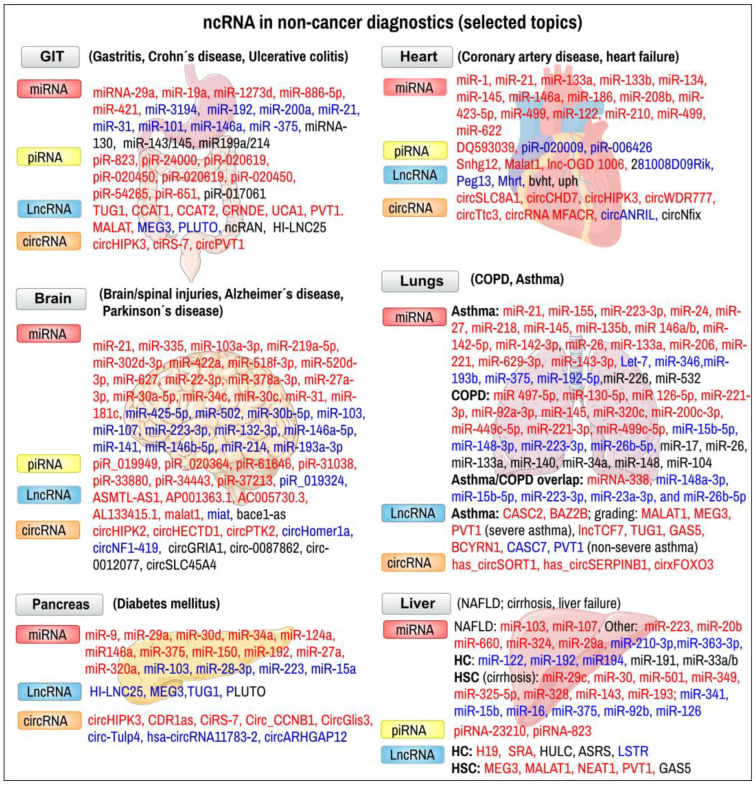
Examples of ncRNAs in non-cancer diagnostic indications. Markers from peripheral blood, tissue (endoscopic biopsy) or other body liquids. In certain markers, their expression levels are indicated by color: red—upregulated markers; blue—downregulated markers. COPD—chronic obstructive pulmonary disease, HC—hepatic cells, HSC—hepatic stellate cells, NAFLD—nonalcoholic fatty liver disease. Certain pictures in the background are credited to Servier Medical Art, France, and used according to the CC-BY-4.0 licence.

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
