# Peer review of "Non-Coding RNAs in Human Cancer and Other Diseases: Overview of the Diagnostic Potential"

_ijms, 2023, doi:10.3390/ijms242216213_

Round 1

Reviewer 1 Report

Comments and Suggestions for Authors

The paper of Dr. Roman Beňačka, Dr. Jozef Radoňak, and co-authors is an impressive documentation on the diagnostic potential of non-coding RNAs.

The paper is well-illustrated, but quotes only 170 references. In addition, 3 figures are out of the scope of the review, because they are illustrations on biogenesis or nomenclature.

The paper may deserve publication in the future, but only after drastically changing the scope of the review, and providing the readers with a logical text giving a clear opinion. Presently, the text ends abruptly at section ‘3. 5. Gastrointestinal diseases’, with a very brief conclusion in ‘5. Conclusions’. The readers would like a conclusion and perspectives from the authors.

I am giving some advises to improve the manuscript, which needs drastic improvements and is close to rejection.

Major modifications

1. Why a review so broad on ‘human cancer and other diseases’?

If you query PubMed on cancer, miRNAs, and diagnostic potential, the number of systematic reviews is over 200, and of primary papers over 5,000.

The authors have to ponder reducing the scope of their paper because anyway, summarizing the broad class of ‘Non-coding RNAs’ is already a very difficult task.

The low number of references quoted indicates that you could even reduce the review to a single type of cancer, for instance.

2. Problem with definition of miRNAs. In addition to the work of 3. Ambros (2001), I would suggest quoting ‘Bartel DP. 2018. Metazoan MicroRNAs. Cell 173: 20–51. doi:10.1016/j.cell.2018.03.006’, which is from the same groupe and more recent.

3. Are Figures 1-3, original illustrations from the authors? If yes, they can be published, but these figures 1-3 cannot replace a clear text on every subtype of non-coding RNAs considered in the scope of the review. A major weak point of the review is that the difficulties in classifying non-coding RNAs are hidden. I can advise reading this point of view on the difficult task of classifying small RNAs: Desvignes, T.; Batzel, P.; Berezikov, E.; Eilbeck, K.; Eppig, J.T.; McAndrews, M.S.; Singer, A.; Postlethwait, J.H. miRNA Nomenclature: A View Incorporating Genetic Origins, Biosynthetic Pathways, and Sequence Variants. Trends Genet. 2015, 31, 613–626.

4. Problem with the numeration of sections, where is section 4 ?

5. Figure 4 is giving useful information on the topic ‘diagnostic potential in cancer’, and Figure 5 on non-cancer diseases. However, the figures 1-3 are not needed.

These figures 4 & 5 are using ‘Pictures in the background with Credit. Servier Medical Ar, France (CC-BY-4.O licence)’, to create the landscape. The figure 5 is poor, not indicating the up or down-regulation of non-coding RNAs used as biomarkers.

My suggestion would be to focus the review on ‘diagnostic potential in cancer’. The sections 3.2 to 3.5 should be removed. If the authors are more familiar with neurological diseases or metabolic diseases, they can choose another focus, but the Figure-5 has to be drastically improved like Figure-4.

5. Nomenclature of miRNAs is erratic and of little use for non-specialists. Most miRNAs have a 3p or 5p forms. For instance, the miR-21 often refers to miR-21-5p, but sometimes to miR-21-3P. The let-7, upper-left of Figure-4, is the family, the information is incomplete.

Minor modifications. The English style of the review is correct. However, minor alterations are of no use here, as the paper should be entirely re-written.

Comments on the Quality of English Language

Minor modifications. The English style of the review is correct. However, minor alterations are of no use here, as the paper should be entirely re-written.

Author Response

Dear reviewer,

Firstly, we would like to express our gratitude for your time, and valuable suggestions.

To answer your 1st comment: We wanted to write review summarizing new information and data regarding non-coding RNAs and their potential to be diagnostic markers for the most common and widely spread diseases. We tried to create the list of the newest ncRNAs which could be suitable indicators in the research and treatment of human diseases in the future.

To answer your 2nd comment: As per your suggestion, we changed the citation 3. Ambros (2001) to Bartel DP. 2018. Metazoan MicroRNAs. Cell 173: 20–51. doi:10.1016/j.cell.2018.03.006.

To answer your 3rd comment: Yes, figures 1 – 3 are original illustrations created by the first author as an appropriate addition to the text.

To answer your 4th comment: We added new section 4: Methodologies in ncRNAs detection. Thank you for notifying us on the problem with numeration of sections on our review.

To answer your 5th comment: According to your suggestion, we edited the Figure 5 according to the Figure 4, indicating changes in regulation (up/down) with appropriate colors. We are grateful for your suggestion. Ultimately, we decided to keep sections 3.2 to 3.5, as per suggestions and positive feedback from other reviewers. We wanted to make a collection of current data regarding ncRNAs in various (possibly not all) common human diseases and the data samples were summarized in Figure 5.

To answer your 6th comment: We corrected all mistakes and misspelling we could. The non-coding RNAs we listed in our review and figures were based on data from particular research papers, that is why the miRNAs are in some cases referred with the addition of -3p/-5p and in some without (e.g. miR-21, miR-142 – miR-142-3p/miR-142-5p).

Minor modifications: We edited the sections 1, 2 and 3 extensively, we revised every sentence that needed to be corrected, including structure, word choice, punctuation and misspellings.

All corrections and changes are highlighted in the text and we hope that all above mentioned modifications will be acceptable and satisfactory.

Reviewer 2 Report

Comments and Suggestions for Authors

This review provides a comprehensive overview of the diverse branches of non-coding RNA, discussing their potential roles as therapeutic targets for various human diseases, including cancer. In my opinion, this review is an excellent source of material for understanding concepts and advances in the biology of non-coding RNA. However, there are multiple instances of grammar and spelling errors in the manuscript. It is essential for the author to thoroughly revise and proofread to rectify these issues.

1.    Verify the accuracy of all diagrams, particularly in Figure 2 and Figure 3. Confirm that the stop codons of pre-mRNA are correctly represented as UAG, UGA, and UAA, not UGG.

2.    Ensure that all legends comprehensively explain the diagrams. For instance, in the legend of Figure 4, the meaning of non-coding RNA marked by the purple color should be explicitly stated for clarity.

3.    Carefully review the listing of non-coding RNAs to ensure accuracy. For instance, in Figure 4, "mi-766" should be corrected to "miR-766."

Comments on the Quality of English Language

1.    Please ensure that all words are spelled correctly and the gramma is used correctly. For instance, in the introduction, the sentence "Recently, the role of this pouzzling part of genome started to be disentangled a the gigantic world of ncRNAs has been siclosed." should be revised to "Recently, the role of this puzzling part of the genome started to be disentangled as the gigantic world of ncRNAs has been disclosed."

2.    Check for consistency in the use of acronyms throughout the review. For example, instead of "not-coding DNA," it may be more appropriate to use "non-coding DNA" for consistency.

Author Response

Dear reviewer,

Firstly, we would like to express our gratitude for your time, and valuable suggestions.

To answer your comments on the quality of English language: we tried to answer all your comments and correct every misspelled words and grammar and revised all sentences that needed to be corrected. We checked the consistency of all acronyms in the text according to your suggestions.

To answer your 1st comment: we verified the accuracy of our diagrams. We corrected the stop codons in the Figure 2 and Figure 3 to UAG, UGA and UAA.

To answer your 2nd comment: we rewrote legends into shorter and more comprehensible versions.

To answer your 3rd comment: we carefully reviewed the list of the non-coding RNAs and corrected all misspellings.

All corrections and changes are highlighted in the text and we hope that all above mentioned modifications will be acceptable and satisfactory.

Reviewer 3 Report

Comments and Suggestions for Authors

This is potentially important overview which discusses the generation and clinical significance of non-coding RNAs. The major concern is the inconsistency in the writing.  Sections 1-2 contain sentences that require extensive editing, including sentence structure, word choice, punctuation, and typos. There are too many issues to list.

In contrast, Section 3 is well-written. Minor comments on this section are listed below.

1. Line 444: The sentence "Piwi-interacting RNAs can act as theraupetic (mis-spelled) agents for breast cancer" should be followed by  relevant citations. Alternatively, the statement can be softened by indicating that these RNAs can potentially act as therapeutic agents. 

2. Line 538: The sentence starting with "Jain et al" needs to be reworded. It is difficult to understand as written.

3. Line 555: There is a typo in the sentence starting with Yan.

4. Line 579:  Is 101 a typo?

5. Line  611: In the sentence starting with "A study by Rajan", delete explored or examined.

6. Sometimes type 2 Diabetes is abbreviated DM2 and sometimes T2DM. It's probably better to use just one of these abbreviations.

7. Line 666: Add an "a" between "play" and "role"

8. Line 671: Substitute "other" for "another".

9. Line 673: Substitute an "a" for the "s" at the end of the line.

10. There is a type in the title of the legend to Figure 4.

Comments on the Quality of English Language

See previous section.

Author Response

Dear reviewer,

Firstly, we would like to express our gratitude for your time, and valuable suggestions.

To answer your comments and suggestions: We edited the section 1 and 2 extensively, we revised every sentence that needed to be corrected, including structure, word choice, punctuation and misspellings. In section 3, we corrected all mistakes you pointed out (1.-10.) and some other corrections to make this review more comprehensive.

All corrections and changes are highlighted in the text and we hope that all above mentioned modifications will be acceptable and satisfactory.

Round 2

Reviewer 3 Report

Comments and Suggestions for Authors

There is still a need for significant editing of the first two sections. There are still many typos, and sentences and paragraphs that need to be rewritten. I’ve listed typos and minor edits that can easily be changed below.

More significant concerns/edits

Line 148 – The sentence starting with “In addition” is difficult to follow and needs to be rewritten. Dividing it into additional sentences would be helpful.

Also, what does “such trials” refer to? 

Line 182 - The sentence starting with “Recent” needs to be rewritten. Specifically it’s not clear what “cumulative data were examining their usage in breast cancer….” means.

Line 199 – It’s unclear what body fluids in urine means.

Line 260 – What does “ss” refer to? If single-stranded, this needs to be spelled out.

Line  280 – What does cell development mean? This should be clarified.

Line 284 – The sentence starting with “Defects” is hard to understand. What does “may lead to include to B+thalassaemia” mean? This sentence needs to be rewritten

Line 390 – The paragraph under Function needs to be completely rewritten. Something like the following. They are involved in the regulation of gene expression both on the transcriptional and epigenetic levels. Additionally, gene regulatory elements are imbedded in lncRNA genes. (These statements need citations) LncRNAs can also act post-transcriptionally by acting as sponges for different mRNAs… Finally LncRNAs can regulate the translation and post-translational modification of proteins in the cytoplasm.

Line 324 – Something is missing before the list following [58].

Line 397 – The punctation and typos in the paragraph starting on this line needs to be edited. The writing is really difficult to follow. For instance, on line 403, the sentence starting with the number 3 is hard to follow.  What does 3 refer to?

Typos and other minor edits

Line 26 - Change encodes to encode.

Line 28 – Change contract to contrast.

Line 35 – Change area to areas.

Line 54 – Add comma after function.

Line 75 – Add and s to product, ie the products of transcription are.

Line 82 – Add a hyphen between single and stranded, ie single-stranded.

Line 95 – Replace the comma after miRNA with a semicolon and add a comma after therefore, ie, suppressor-miRNA; therefore, the….

Line 151- Applications is mis-spelled.

Line 160 – Change high to highly.

Line 195 – Add “the” before miRNA sponge pathway

Line 222 – Sentence starting with “Two PIWI proteins”  should be edited. Add a period after PIWI2. Start a new sentence with Overexpression.

Line 239 – Delete “the” before embryogenesis.

Line 240 – The sentence starting “Transposable elements is awkward.  How about the following.  Transposable elements risk damaging the genome through their transposition; therefore, …

Line 254 – Add a space between RNA and is.

Line 255 – Add an s to plays, ie RNA group plays. Maybe change “parts” to “roles” and “an assembly” to “the assembly”

Line 256 - Add a period after) and start a new sentence with “These ribonucleoprotein complexes perform the precise removal….  Delete “inevitable to”

Line 257 – Delete e from exones, ie, exons.

Line 268 – Add a comma after sequence characteristics. Delete “and” before protein cofactors and add a comma and the an and before, snRNA uses. How about - Based on shared sequence characteristics, protein cofactors, and functions, snRNAs can be divided into…. ?

Line 270 – Change “the use” to “their use,” and add a comma after use.

Line 300 – Add a space between ) and are

Line 318 – Add a space between Function. and snoRNAs

Line 320 – Add “such “before “as” to read such as.

Line 331 – Imprinting is miss-spelled

Line 338 – Change to “snoRNAs have been…” or “snoRNA has been..”  There is a typo as written, ie have have been.

Line 343 – Add a space between “SNORD113” and “are”

Line 325 – Change “In opposite” to “In contrast”

Line 370 – Add a space between ) and is.

Line 371 – Add an “a” before mysterious.

Line 374 – Add a space between ) and is.

Comments on the Quality of English Language

See comment to the authors

Author Response

Dear reviewer, we would like to again express our gratitude for your time and suggestions. We have revised and corrected all your major and minor concerns and additional mistakes we have noticed.All corrections and changes are highlighted in the text and we hope that all of the changes mentioned above will be acceptable and satisfactory.

Round 3

Reviewer 3 Report

Comments and Suggestions for Authors

This manuscript is greatly improved.

Minor typos

Pg 2 line 3- Delete of before several

Pg2 line line 8 – Add a space between ncRNAs and are.

Pg6 line 3. The sentence is preceded with (b). However there is no (a) listed in the section “Clinical applications” Maybe the (b) was carried from a previous version?

Comments on the Quality of English Language

The writing is significantly improved.

Author Response

Dear reviewer, we are glad to hear our manuscript is greatly improved, we have corrected both typos on page 2 (line 3 and line 8). Regarding page 6 line 3, there is a) preceding b) on page 5 line 45, we have highlighted the a) in the text along with other changes. We are very grateful for your valuable time and suggestions.